# Compression-Softening Bond Model for Non-Water Reactive Foaming Polyurethane Grouting Material

**DOI:** 10.3390/polym15061493

**Published:** 2023-03-16

**Authors:** Boyuan Dong, Mingrui Du, Hongyuan Fang, Fuming Wang, Haoyue Zhang, Longhui Zhu

**Affiliations:** 1School of Water Conservancy and Civil Engineering, Zhengzhou University, Zhengzhou 450001, China; 2National Local Joint Engineering Laboratory of Major Infrastructure Testing and Rehabilitation Technology, Zhengzhou 450001, China; 3Collaborative Innovation Center of Water Conservancy and Transportation Infrastructure Safety, Zhengzhou 450001, China; 4Yellow River Laboratory, Zhengzhou University, Zhengzhou 450001, China; 5Shenzhen Feiyang Protech Corp., Ltd., Shenzhen 518000, China

**Keywords:** discrete element, elastic-brittle-plastic assumption, micro-foams, non-water reactive foaming polyurethane grouting material

## Abstract

In this study, the uniaxial compression and cyclic loading and unloading experiments were conducted on the non-water reactive foaming polyurethane (NRFP) grouting material with a density of 0.29 g/cm^3^, and the microstructure was characterized using scanning electron microscope (SEM) method. Based on the uniaxial compression and SEM characterization results and the elastic-brittle-plastic assumption, a compression softening bond (CSB) model describing the mechanical behavior of micro-foam walls under compression was proposed, and it was assigned to the particle units in a particle flow code (PFC) model simulating the NRFP sample. Results show that the NRFP grouting materials are porous mediums consisting of numerous micro-foams, and with the increasing density, the diameter of the micro-foams increases and the micro-foam walls become thicker. Under compression, the micro-foam walls crack, and the cracks are mainly perpendicular to the loading direction. The compressive stress–strain curve of the NRFP sample contains the linear increasing stage, yielding stage, yield plateau stage, and strain hardening stage, and the compressive strength and elastic modulus are 5.72 MPa and 83.2 MPa, respectively. Under the cyclic loading and unloading, when the number of cycles increases, the residual strain increases, and there is little difference between the modulus during the loading and unloading processes. The stress–strain curves of the PFC model under uniaxial compression and cyclic loading and unloading are consistent with the experimental ones, well indicating the feasibility of using the CSB model and PFC simulation method to study the mechanical properties of NRFP grouting materials. The failure of the contact elements in the simulation model causes the yielding of the sample. The yield deformation propagates almost perpendicular to the loading direction and is distributed in the material layer by layer, which ultimately results in the bulging deformation of the sample. This paper provides a new insight into the application of the discrete element numerical method in NRFP grouting materials.

## 1. Introduction

Non-water reactive foaming polyurethane (NRFP) grouting material refers to the product of the polymerization reaction between isocyanates and polyols, and it has become one of the most commonly used materials in polymer grouting projects in civil engineering [1,2]. The two raw materials are injected into the soil cavities or rock fissures under high pressure for the reinforcing purpose, and the polymerization reaction can be completed in a matter of ten minutes, and the NRFP matrices with expanded volume and strong impermeability are formed [3,4,5]. It has been reported that the NRFP grouting material has an expansion volume ratio of 10–20 and a permeability coefficient of 10^−8^ cm/s, and it is environmentally friendly [6]. By virtue of these superior material properties, NRFP grouting material has found its applications in the non-excavation rehabilitation of expressway pavement [7], underground pipes [8], and foundations [9,10], as well as building water barriers.

As shown in Figure 1, in practical applications, NRFP sustains the external load all the time, and its mechanical behavior is critical for the long-term reinforcing effect. This is because the failure of the NRFP matrices can cause the re-failure of the enhanced engineering structures. Many studies have been conducted to understand the mechanical properties of NRFP grouting material. For example, Xiang et al., have experimentally studied the failure characteristics of NRFP grouting material under the fatigue load [11,12]; Fang et al., have studied the effect of loading rate on the uniaxial compression mechanical properties of NRFP [13,14]. In another study, Fang et al. established the constitutive model for NRFP grouting material based on the uniaxial compressive experimental results. Li et al. studied the effect of geometry size on the compressive mechanical properties of NRFP [15]. Wang et al. investigated the influences of different chemical corrosion environments on the compression strength performance of NRFP grouting material [16]. Liu et al. analyzed the failure mode and mechanical properties of polyurethane composites by combining macro and micro analysis [17]. NRFP grouting material is a typical loose porous matrix and it performs with the elastic-plastic characteristic [18,19]. It has been reported that the macroscopic deformation and failure of NRFP grouting material are caused by the wall of the damage of the micro-foams. From this perspective, it is necessary to study the macroscopic mechanical behavior of porous NRFP grouting material by investigating the mechanical behavior of micro-foams, which, however, is overlooked. This is because the size of the micro-foams in the NRFP grouting material is on the micron scale, and it is difficult to directly observe the mechanical behavior and failure process of the micro-foams with the current non-destructive testing and monitoring methods. The numerical simulation method can be an economical and effective auxiliary method to understand the micron-scale mechanical behavior of NRFP grouting material. Some researchers have proposed different constitutive models for foamed polymer materials and studied the mechanical properties of polymers with the finite element method based on the assumption that NRFP grouting material is a continuous medium [20,21,22]. The molecular dynamics simulation method has also been used to study the microscopic scale tensile mechanical response property of the pure NRFP grouting material [23], but not the micro-scaled micro-foams.

The discrete element method (DEM) is a particle-based numerical simulation method [24]. The computational domain of DEM is not a continuous medium. Instead, it is discretized into interacting particle ensembles (two-dimensional disk (2D) and three-dimensional sphere (3D) are the main particle shape in space) by constructing particle ensembles into specimens of different geometries, utilizing a contact model forming interactions and iterative analysis based on Newton’s second law to make the macroscopic mechanical properties of the numerical specimen approach those of real materials [25]. Therefore, with the DEM method, the problems of discontinuities, high gradient variables, and remeshing in fracture or damage can be overcome [26,27,28,29]. The DEM method has been widely used to describe the macroscopic mechanical properties of materials, such as rock [30,31], soil [32], and concrete [33]. The accuracy of the DEM simulation results mainly depends on the contact model of the particles, and different contact models have been proposed to be suitable for different materials. For example, Potyondy and Cundall proposed a BPM model with a linear contact force dual spring damper to reproduce many features of rock behavior, including elasticity, fracturing, acoustic emission, damage accumulation, producing material anisotropy and an increase in strength with confinement [34]. Based on the Hertz contact theory, Walton and Thornton developed a nonlinear contact model that reflects mechanical behavior between particles of the ideal smooth sphere [35,36]. Kasyap et al. attempted to define the DEM contact model based on the microscopic experimental results of reproduced flat LBS grains with artificial bonds [37]. Ma et al. established a contact model that contains the tensile softening behavior of quasi-brittle materials, and this model can be used to describe the tensile-compression strength ratio more accurately [38]. All these studies provide references for the study of the mechanical properties of NRFP grouting material based on the DEM method. In this study, based on the scanning electron microscope (SEM) observation result, the uniaxial compression experimental result, and the assumption of elastic-brittle-plastic, the compression-softening bond (CSB) model was proposed for NRFP grouting material, followed by the simulation with the three-dimensional particle flow code (PFC) method, one of the commonly used DEM methods [39]. The results show that the CSB model established here can describe the mechanical behavior of NRFP grouting material under the uniaxial compression and cyclic loading and unloading well, which highlights a new idea to study the mechanical properties of NRFP grouting material with the PFC simulation method.

## 2. Experiment Procedure and Results

### 2.1. Raw Materials and Samples Preparation

In practical engineering, when NRFP grouting material has been injected into geological defects, the main factor that affects the enhancing performance is the density. This is because both the mechanical properties and volume expansion rate of NRFP grouting material increase with increasing density. The density of the currently used NRFP grouting material is generally lower than 0.5 g/cm^3^. In this study, the specimen with a density of 0.29 g/cm^3^ was prepared. As shown in Figure 2, the self-developed high-pressure grouting equipment was used to prepare the NRFP samples. The commercially available isocyanates and polyols liquids provided by Wanhua Chemical Group Co., Ltd. (Shandong, China), were injected into the cube-closed molds with a side length of 70.7 mm through two different pipes under the nitrogen gas pressure of about 8.0 MPa. Under this pressure, the two liquids could be atomized into microdroplets, and thus the two raw materials were fully and uniformly mixed to produce the NRFP samples that are as homogeneous as possible. The isocyanate and polyol were mixed with a mass ratio of 1:1. After grouting, the samples were cured in the environment with a temperature of 25 °C and relative humidity of about 40% for 1 h before demolding, followed by curing for another two hours in the same environment. The prepared NRFP samples were cut into small cubes with a side length of 10 mm for the subsequent uniaxial compression and cyclic loading/unloading experiments.

### 2.2. Uniaxial Compression and Cyclic Loading-Unloading Experiments

The UTM-2203 type microcomputer-controlled electronic universal testing machine (as shown in Figure 3) was used to conduct the uniaxial compression and cyclic loading-unloading experiments. The displacement loading method at the rate of 0.05 mm/min was used to apply the axial load. The main purpose of conducting the cyclic loading-unloading compression experiment is to verify whether the contact model proposed here can describe the mechanical response of NRFP grouting material after yielding. Therefore, when conducting the cyclic loading-unloading compression experiment, the samples were loaded to yield, then unloaded to half the yield stress, and then loaded to yield stress again, and so on many times until the maximum strain reached 0.15. Both the loading and unloading processes were carried out at the rate of 0.05 mm/min.

### 2.3. Scanning Electron Microscope Characterization

The scanning electron microscope (SEM) characterizations on the micromorphology of NRFP samples was conducted using a 200FEG-type SEM testing system. SEM characterizations were also conducted on the samples with other different densities (0.26 g/cm^3^, 0.46 g/cm^3^, and 0.68 g/cm^3^) for investigating the influence of density on micromorphology. The observed area on each cross-section was randomly chosen. The micromorphology of the samples was observed at an accelerating voltage of approximately 25.0 kV and amplification factors of 100–200. To improve the samples’ conductivity, the gold spraying device is used to spray gold on the target section before observation.

### 2.4. Experimental Results

Figure 4a shows the microstructure of NRFP grouting material with different densities. At the micro-scale, the NRFP grouting material is loose and porous, and numerous closed micro-foams can be observed. The micro-foams are approximately spherical, with a diameter of about 10^1^–10^2^ μm, and their diameter decrease with the increase in density. NRFP grouting materials with a lower density have higher porosity and more tightly arranged micro-foams, and thus they can be classified as the closed-cell porous materials. When the density increases further, the micro-foam volume of the material continuously declines, the micro-foam walls become thicker and thicker, gradually developing the elasticity of polyurethane [13]. Previous studies have pointed out that when the NRFP grouting material is dense enough, it is no longer classified as a closed-cell porous material but undergoes brittle fracture under loading [13,15,16]. Figure 4b shows the SEM image of the sample with a density of 0.29 g/cm^3^ after loading. It can be found that there exist discrete cracks on the micro-foams wall, and the direction of these cracks is mostly perpendicular to the loading direction.

The representative stress–strain curve of the NRFP sample under uniaxial compression is shown in Figure 5. From Figure 5, it can be seen that in the initial loading stage (ab section), the axial stress increases almost linearly with the increase of the strain. In this stage, the extrusion between adjacent micro-foams does not cause their walls to be damaged, and the internal deformation that occurs in this stage could be fully recovered after unloading, which is known as elastic deformation. After the b point at which the strain is approximately 5%, the stress–strain curve begins to gradually deviate from linearity (bc section). This is because some micro-foams in the sample ruptured and plastic deformation occurred, which results in decreased stiffness. After yielding, a plateau can be observed on the stress–strain curve (cd section) and, in this stage, with the furtherly increased axial strain, the axial stress remains almost unchanged. The axial stress starts to increase again when the strain reaches a certain value (de section), indicating that strain-hardening behavior occurs [14]. According to Figure 5, it can be calculated that the yield strength and elastic modulus of the NRFP sample are approximately 5.72 MPa and 83.2 MPa, respectively.

The stress–strain curve of the NRFP sample under the cyclic loading-unloading process is shown in Figure 6a. From Figure 6c, it can be seen that before the strain-hardening stage, the stress–strain curve is almost consistent with the uniaxial compression stress–strain curve. The yield strength and elastic modulus of this sample are approximately 5.84 MPa and 83.5 MPa, respectively. From Figure 6b,d, it can be seen that with the increase in the number of cycles, the residual strain *ε_n_* increases but its increment decreases, and there is little difference between the modulus of the material in the loading and unloading processes. For example, when the number of cycles is 1, *ε_n_* is approximately 1.08 %, and when the number of cycles is 9, *ε_n_* is approximately 1.19 %. During the loading and unloading processes, due to the continuous deformation and failure of the internal cellular structures of the material and the creep effect, the plastic strain of the sample increases when the number of cycles increases.

## 3. Establishment of CSB Model and Numerical Simulation Details

The SEM characterization results show that NRFP grouting material contains numerous micro-foams, and it is the damage and failure of the micro-foams wall that cause the failure of the sample (as shown in Figure 4b). According to the theory of the DEM method, here, the micro-foams in the NRFP grouting material are mapped as the balls in the DEM model, and the damage of the micro-foam wall can be considered as the failure of the contact interaction between each ball. This mapping assumption is shown in Figure 7. With this assumption, the huge number of balls that are required to constitute the DEM model simulating the NRFP grouting material is significantly declined, and thus the computational efficiency can be improved. A contact model that can well describe the mechanical behavior of micro-foam walls is the key to obtain reasonable simulation results.

Previous researchers have proposed the linear-elastic discrete element model to describe the mechanical interactions between the balls simulating the brittle or quasi-brittle porous media materials such as rocks and concrete [40,41]. With this contact model, the balls are assumed to be rigid and there is no damage to them under compression or tension. However, for the hollow micro-foams in NRFP grouting material, they could be compressed, and their curved walls indicate that they may lose part of the bearing capacity. Therefore, the mechanical interactions between each micro-foam do not follow the linear-elastic assumption and the micro-foams are not damage-free when they move toward each other. To directly observe the micro-mechanical response of micro-foams of NRFP through experiments is not possible. Therefore, in this study, based on the experimental results presented in Section 2.4, the trial-and-error method was adopted to describe the nonlinear mechanical response between the micro-foams as well as possible.

Three contact models that conform to the elastic assumption, elastic-brittle-plastic assumption, and elastic-brittle assumption are selected for numerical unit experiments in PFC during micro-foams compression. The results are shown in Figure 8. It can be seen that when the contact model based on the elastic assumption is used, with the increasing strain, the axial stress increases gradually to the peak value, and then it declines rapidly after having reached the yield strain. As for the numerical stress–strain curve obtained based on the contact model that follows the elastic-plastic assumption, it contains two stages, but does not contain the yield plateau. Neither the stress–strain curve obtained based on the contact model following the elastic assumption nor the one obtained based on the contact model following the elastic-plastic assumption is consistent with the experimental one (as shown in Figure 5). The stress–strain curve obtained based on the elastic-brittle-plastic assumption contains both the linear growth stage and the yield plateau stage, and it is more consistent with the experimentally obtained stress–strain curve of the NRFP grouting material. This model considers the ball overlap as the control condition for the yield of the contact element, which confirms the failure characteristics of the micro-foams in NRFP grouting material. Therefore, the compression-softening contact model based on the elastic-brittle-plastic assumption is established.

### 3.1. Compression-Softening Bond Model Based on the Elastic-Brittle-Plastic Assumption

The compression-softening bond (CSB) model is proposed based on the linear contact bond (LCB) model because this study mainly concerns the mechanical response of the specimen under compression [34]. As shown in Figure 9, the CSB model can be considered as adding two springs with constant stiffness at the bond point (*k_n_* and *k_s_*). These two spring components have specified tensile strength and *k_n_* is connected in series to a bonded slider with specified bond strength. The bonded slider plays the role of a plain slider when the normal force does not reach the threshold, but when the force exceeds the threshold, its bond strength declines to a certain value. Specifically, according to the elastic-brittle-plastic assumption, CSB can be described as follows: when the two balls in the numerical models move towards each other, $F_{n}^{l}$, the normal force is initially linear elasticity, and the yield condition of the cell is described by the degree of deformation, which is described by the yield strain coefficient $\varepsilon_{ysc}$. When the surface gap $g_{s}$ of the two balls reaches $\varepsilon_{ysc}$, the contact model begins to yield, and the normal force suddenly drops from ${\left(F_{n}^{l}\right)}_{ps}$ to $RSC\times {\left(F_{n}^{l}\right)}_{ps}$. *RSC* (0 < *RSC* ≤ 1), the residual stress coefficient, is a parameter that reflects the softening degree of the material, and it represents the axial bearing capacity of the micro-foams after yielding. *RSC* is defined because the SEM characterization results in Section 2.4 show that the micro-foams remain connected to each other even after damaging, and they can still be compressed. Therefore, after yielding, the residual part of the micro-foams is still with some load-bearing capacity. Besides, the nonlinear mechanical response of micro-foams after yielding is complex, which is assumed as brittle-plasticity here, and the residual load-bearing capacity is defined as a constant that depends on *RSC*. The larger the *RSC* value, the greater the residual load-bearing capacity of the micro-foams after yielding.

The contact points always perform with the linearly elastic behavior during the separated motion, and for the contact elements that have yielded already, in the reloading process, plastic yield occurs directly when the stress reaches $RSC\times {\left(F_{n}^{l}\right)}_{ps}$. When the normal force exceeds the tensile strength $\left(F_{n}^{l}>T_{F}\right)$ during the separated motion, the bond breaks, and both the normal and shear forces are set to zero $\left(F_{n}^{l}>T_{F}\right)$. The specific force-displacement criterion is shown in Equation (1) (as shown in Figure 10a).

The linear shear force is updated as Equation (2) (Figure 10b). The existence of the contact bond precludes the possibility of slippage. Thus, if the shear force exceeds the shear strength $\left(∥F_{s}^{l}∥>S_{F}\right)$, the bond breaks $\left(F_{n}^{l}=0\;if\;F_{n}^{l}>0\right)$, but the contact force remains unchanged, provided that the shear force does not exceed the product of the coefficient of friction and the normal force, and the normal force is compressive.
(1)$$\begin{array}{l} F_{n}^{l}=\left\{\begin{array}{c} {\left(F_{n}^{l}\right)}_{0}+k_{n}\Delta \delta_{n},-g_{s}<\varepsilon_{ysc}\left(R^{\left(1\right)}+R^{\left(2\right)}\right)\\ RSC\times {\left(F_{n}^{l}\right)}_{ps},-g_{s}\geq \varepsilon_{ysc}\left(R^{\left(1\right)}+R^{\left(2\right)}\right) \end{array}\right.,\\ {\left(F_{n}^{l}\right)}_{ps}=\sum_{-g_{s}<\varepsilon_{ysc}\left(R^{\left(1\right)}+R^{\left(2\right)}\right)}k_{n}\Delta \delta_{n} \end{array}$$
where ${\left(F_{n}^{l}\right)}_{0}$ is the normal force at the previous moment, $k_{n}$ is normal stiffness, $\Delta \delta_{n}$ is the normal displacement from the previous moment to the current moment, $g_{s}$ is the overlap of two balls connected through the contact element, $R^{\left(1\right)}$ and $R^{\left(2\right)}$ is the radius of the two balls connected through the contact element, *RSC* is residual stress coefficient, ${\left(F_{n}^{l}\right)}_{ps}$ is the yield stress controlled by $\varepsilon_{ysc}$.
(2)$$F_{s}^{l}={\left(F_{s}^{l}\right)}_{0}-k_{s}\Delta \delta_{s}$$
where ${\left(F_{s}^{l}\right)}_{0}$ is the shear force at the previous moment, $k_{s}$ is shear stiffness, and $\Delta \delta_{s}$ is the shear displacement from the previous moment to the current moment.

### 3.2. Verification of the CSB Model

To verify the feasibility of the CSB model, as shown in Figure 11a, a simple PFC model consisting of two ball elements with radius of 0.5 was established for the compression simulation. The compression simulation was achieved by fixing the bottom ball element but moving the upper one at a constant speed of 0.1/s. The CSB model was used to describe the contact interaction between two ball elements, and the normal stiffness, yield strain coefficient $\varepsilon_{ysc}$, and residual stress coefficient *RSC* are defined as 20, 0.1, and 0.3, respectively. In this simulation, it was assumed that the yielding occurred when the overlap degree reached 10% of the sum of the ball unit radii. The unloading process began when the overlap reached 20% of the radius of the balls, and it stopped when the overlap declined to 0. Then, the loading was carried out again until the overlap reached 30% of the radius of the balls. During all these processes, the normal force of the contact elements between the two balls was monitored.

Figure 11b shows the variation of normal force of the contact element during the loading, unloading, and re-loading processes. From Figure 11b, it can be seen that the normal force increases almost linearly when the overlap increases, it declines to the residual strength when overlap reaches 10%, and it declines to 0 when the overlap declines to 0. During the reloading process, the plastic yield occurs after the normal force $F_{n}^{l}$ reaches the yield stress. The variation of the normal force of the simulation model is consistent with the assumption in Section 3.1, indicating that the CSB model proposed can be used to describe the mechanical behavior of micro-foams in NRFP grouting material under compression.

### 3.3. Establishment of the PFC Model Simulating the NRFP Samples

As shown in Figure 12, a closed cube space with a side length of 10 mm was built, and it is enclosed by the wall units. Then, this closed space was filled with numerous randomly distributed balls of which the radius is 0.95 *R*, followed by enlarging the radius of the balls to *R* to fill the confined space to the maximum extent and minimize the unbalanced force within the model. Referring to the actual size of the micro-foams in the NRFP grouting material presented in Figure 4b, R_0_, the basic radius of the balls in the model, was taken as 100–125 microns (as shown in Table 1).

The CSB model proposed in Section 3.1 was assigned to the contact interactions between each ball, and the values of different parameters in the CSB model are listed in Table 2. The process of determining the values of modulus, stiffness ratio, tensile strength, and shear strength is detailed in previous publications [42]. The yield strain coefficient and *RSC* were initially assigned different values and their specific values were finally determined according to the calibration process, as shown in Figure 13.

After having determined the values of yield strain coefficient and *RSC*, both the uniaxial compression and cyclic loading and unloading compression were simulated with the PFC model. The compression loading in PFC simulations is often achieved by moving the wall unit that acts as the loading platform. Strain controls loading strategy and the stress control loading strategy can be realized using servo control. In this study, the compression simulation is achieved by applying a vertical velocity to the wall unit on the upper surface while fixing the wall unit on the lower surface. The velocity of the wall unit on the upper surface was 0.1 mm/min. Referring to the cyclic loading and unloading simulation, in this study, the PFC model was initially subjected the uniaxial compression until the strain reached 13.8%, followed by altering the loading strategy to cyclic loading and unloading. The stress waveform under cyclic loading and unloading was represented by a continuous cosine equation (as shown in Figure 14). The peak stress S_max_ was the stress corresponding to 13.8% of the strain, and the minimum stress S_min_ was half the S_max_. The stress waveform is a little different from the experimental one but similar.

## 4. Simulation Results and Discussion

### 4.1. Uniaxial Compression Simulation Results

The radius of the balls in the PFC model has a great influence on the number of balls and the number of contact points. Figure 15a shows the compressive stress–strain curves of the PFC models consisting of balls with different radius: 1.0 × R_0_, 1.5 × R_0_, 2.0 × R_0_, and 3.0 × R_0_. From Figure 15a, it can be seen that as the radius of the balls increases, the fluctuation of the stress–strain curve gradually increased after entering the plateau stage, in which the plastic yielding occurs, and the yield strength increased. This is because the reduction in the number of balls and number of contact elements can reduce the uniformity of the model, and thus the fluctuation of the stress–strain curves is increased. From Figure 15a, it can also be found that the maximum increment in yield strength caused by the increased R_0_ is approximately 0.7 MPa, implying that the varied R_0_ has little influence on the mechanical behavior of the PFC model under compression.

In Figure 15b, the impact of different yield strain coefficient on the mechanical behavior of the specimen is demonstrated. From Figure 15b, it can be seen that the compressive yield strength of the models increases gradually with the increase of $\varepsilon_{ysc}$, and the modulus in the yield plateau stage also slightly increases. This is because after having exceeded the yield threshold, some contact elements undergo the reduction in their load-bearing capacity. However, as a whole, the stress in the numerical specimen gradually propagates to the surrounding contact elements. As the number of yielding contact elements increases, material damage starts to expand at a relatively constant rate, with no significant changes in stress across the entire specimen. By increasing the yield strain coefficient, the onset of this phenomenon can be delayed, increasing both the yield strain and compressive strength. In addition, the contact elements in the whole model are less likely to enter the yield state when the yield strain coefficient increases. Consequently, there is a slight increase in the modulus when the stress-strain curve is in the yield plateau stage.

Figure 15c demonstrates the influence of *RSC* on the stress-strain curve of the specimen. From this figure, it can be seen that when *RSC* is equal to 0, the stress-strain curve contains the stress drop section after the stress plateau stage; when *RCS* is equal to 0.1, the axial stress keeps almost unchanged after yielding, and the stress plateau can be seen on the stress-strain curve; when *RSC* is greater than or equal to 0.2, after yielding, the axial stress also shows an increasing trend after yielding, but with a significantly declined slope. This observation shows that as the *RSC* increases, the modulus of the specimen after yielding increases. This is because the increased residual load-bearing capacity leads to an increase in the stability of the whole system, which can reduce the efficiency of stress transmission and thus reduce the possibility of micro-foams yielding in the material.

Each parameter in the numerical model affects another, so the selection of parameters takes into account the volatility, platform modulus, and computational efficiency. The radius, $\varepsilon_{ysc}$, and *RSC* are 1.5 × R_0_, 0.675, and 0.1, respectively.

Figure 16 displays the comparison of the experimental and simulation stress–strain curves. From Figure 16, it can be seen that the stress–strain curve obtained with the PFC simulation method also contains the linear increasing state, yielding state, and yield plateau stage, which is similar to the experimental stress–strain curve. The yield strength and elastic modulus of the PFC model are 5.81 MPa and 86.2 MPa, respectively, which are close to the experimental results. The changing rate of modulus at the inflection point of the curve from the linear elastic stage to the yielding stage is higher than that of the laboratory data. The reason for this discrepancy is that the mechanical parameters between the contact elements in the PFC model are uniform, whereas the mechanical properties of the micro-foams in the real NRFP sample are different and vary, especially in the yielding stage. In general, when the reasonable parameters and boundary conditions are used, the PFC simulation results based on the CSB model correspond well with the laboratory testing results.

Figure 16 also presents the correlation between the number of the yield contact elements and the axial strain. It is apparent that there are no yield contact elements in the linear elastic stage, and after yielding, more and more contact elements begin to yield as the strain increases, and there is no overall structural damage to the material. Unlike brittle materials, the failure of which has been characterized as contact fracture in DEM, the damage of micro-units in the CSB model is mainly due to the accumulation of yielding micro-foams in space and quantity. To investigate the distribution characteristics of the yield damage under compression, the spatial distribution of yield contact elements at different axial strain and the displacement cloud image of the particles were recorded, as shown in Figure 17. From Figure 17, it can be seen that the spatial distribution of the yield contact elements in the model often appeared from the middle part of the model and gradually expanded to the boundaries. The yield contact elements are stacked together in layers. Referring to the displacement cloud image of the specimen, the particles near the middle part of the model are distributed horizontally and outward during compression, resulting in bulging deformation.

In the uniaxial compression experiment of the specimen, the spatial distribution of the yielded contact is disordered. To reveal the damage evolution of the specimen, based on Figure 18a, a spherical coordinate is defined, and according to the angle with the positive z-axis (*φ*), it is divided into six parts: 1 (0–33.6°, 146.4–180°), 2 (33.6–48.2°, 131.8–146.4°), 3 (48.2–60°, 120–131.8°), 4 (60–70.5°, 109.5–120°), 5 (70.5–80.4°, 99.6–109.5°), 6 (80.4–99.6°). This dividing method can ensure that each part has the same surface area of a sphere of unit radius. Then, *α*, the angle in the normal direction of the contact, needs to be calculated in the same way, and the qualified contacts should be included in each part for statistics. A histogram of the number of contacts in each part before and after loading is generated and it is included in Figure 18a. It can be seen that the proportion of small-angle contacts after loading increases, indicating that the contact direction is concentrated near the loading direction.

Using the same principle, the equally divided spherical area in space is divided into 256 radial pyramids according to the angle as the statistical interval. If the normal direction of contact falls within the range of the direction represented by a pyramid, it will be included in this pyramid statistic. The height and color of each pyramid change with the statistical quantity. The higher the pyramid, the warmer the color, and the more concentrated the distribution. The higher the cone, the warmer the color, and the more concentrated the distribution. The normal direction data of contacts before and after loading in PFC were exported, generating a three-dimensional statistical chart, as shown in Figure 18b,c. Figure 18b,c shows that in the initial stage, the distribution of the normal direction of contacts in the numerical specimen is spherical, indicating that the distribution of the contact direction is uniform. After loading, the shape of the statistical graph of the normal direction of contacts is more like a peanut, which intuitively shows the tendency of the contact direction to concentrate in the loading direction. In combination with the spatial distribution of the yielding contact in Figure 17, it can be concluded that, under compression, the failure of the material is characterized as layered superposition failure and horizontal bulging deformation, which is similar to the experimental results [15].

### 4.2. Cyclic Loading Simulation Results

The loading and unloading modulus between contacts defined by the CSB model are consistent. Figure 19 shows the comparison results of stress–strain curves of the NRFP sample obtained from numerical simulation and laboratory tests. The mechanical properties of the simulation model, including the loading and unloading modulus and yield strength, conform to the experimental results. According to the simulation results, the plastic damage of the material mainly occurs when the axial stress reaches the maximum in each cycle. However, the area of the plastic hysteresis loop in the simulation result is smaller than the experimental result, and more cycles are needed to achieve the same plastic damage result. The main reasons for this difference are that the micro-foams in real materials are not strictly hollow spherical shells, and the wall thickness is not uniform. Moreover, during cyclic loading-unloading process, the nonlinear mechanical response of micro-foams is complex. However, the elastic-brittle-plastic assumption also simplifies the actual mechanical response of micro-foams.

Figure 20 shows the variations of residual strain and residual strain increment of the NRFP sample with the number of loading and unloading cycles. The number of loading and unloading cycles was normalized. From Figure 20, it can be seen that with the increasing number of loading and unloading cycles, the residual strain increases whereas its increment declines. The numerical simulation results are consistent with the experimental results. The damaging process of the materials during cyclic loading and unloading is as follows: the micro-foam walls containing initial damage yield under load, changing the spatial position of the micro-foams and the stress conduction, which can make other micro-foams yield. When the number of loading and unloading cycles increase, more and more micro-foams containing the initial damage yielded, and the spatial distribution of the micro-foams tended to stabilize, resulting in the gradual reduction of the newly yielded micro-foams.

## 5. Conclusions

In this study, based on the experimental results and the elastic-brittle-plastic assumption, a compression-softening bond (CSB) model describing the contact interaction of the DEM model for NRFP grouting material was proposed. The feasibility of the CSB model was verified by comparing the simulation results of uniaxial compression and cyclic loading-unloading compression to the experimental ones. The following conclusions are drawn:(1)NRFP grouting materials are porous mediums consisting of numerous micro-foams with approximately spherical shape. With the increasing density, the diameter of the micro-foams in the NRFP grouting materials declines, and the micro-foam walls become thicker. The micro-foams in the NRFP grouting material with a density of 0.29 g/cm^3^ have the diameter range of 10^1^–10^2^ μm. Under compression, the micro-foam walls crack, and the cracks are mainly perpendicular to the loading direction.(2)The compressive stress–strain curve of the NRFP sample with a density of 0.29 g/cm^3^ contains a linear increasing stage, yielding stage, yield plateau stage, and strain hardening stage. The compressive strength and elastic modulus of the sample are 5.72 MPa and 83.2 MPa, respectively. Under the cyclic loading and unloading condition, when the number of cycles increase, the residual strain increases, but its increment decreases, and there is little difference between the modulus in the loading and unloading processes.(3)The CSB model proposed here can be used to describe the mechanical behavior of NRFP grouting material under uniaxial compression and cyclic compression with the PFC simulation method. Both the stress–strain curves of the simulation model under uniaxial compression and cyclic loading and unloading are consistent with the experimental ones. The compressive yield strength and elastic modulus of the simulation model are approximately 5.81 MPa and 86.2 MPa, respectively.(4)The failure of the contact elements in the simulation model causes the yielding of the sample, and the yield plateau on the stress–strain curve. The yield deformation propagates almost perpendicular to the loading direction and is distributed in the material layer by layer, which ultimately results in the bulging deformation of the sample.

## Figures and Tables

**Figure 1 polymers-15-01493-f001:**
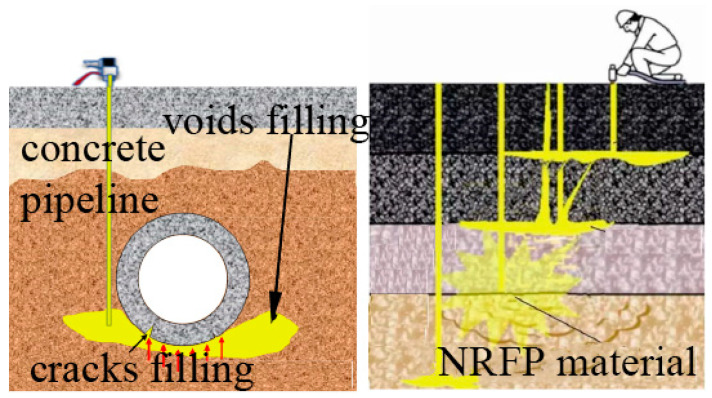
Application of polymer grouting technology in civil engineering.

**Figure 2 polymers-15-01493-f002:**
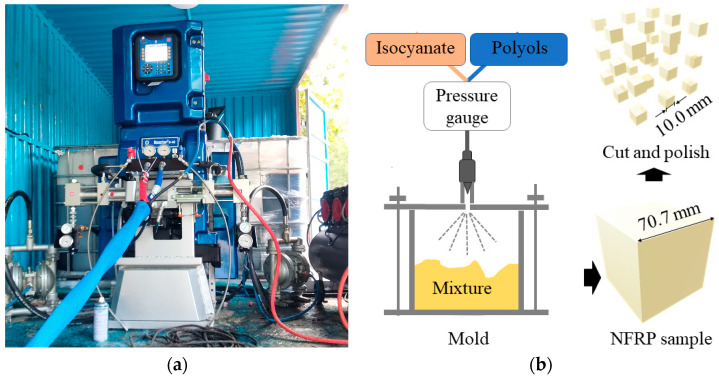
(**a**) The self-developed high-pressure grouting equipment and (**b**) the preparation process of the NRFP samples.

**Figure 3 polymers-15-01493-f003:**
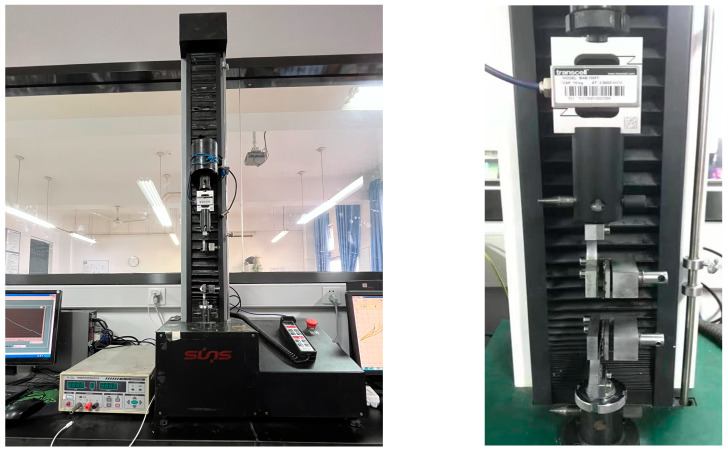
Electronic universal testing machine.

**Figure 4 polymers-15-01493-f004:**
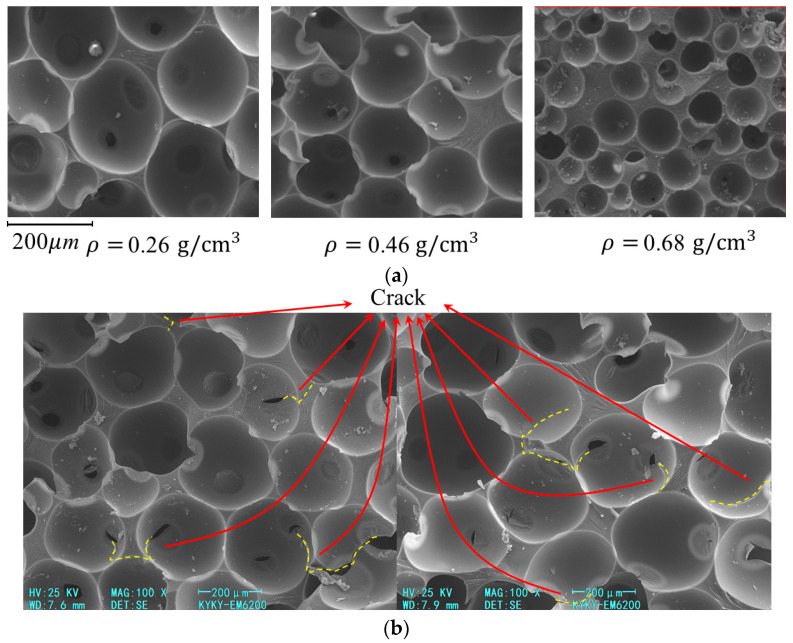
(**a**) SEM images of polymers with different densities and (**b**) SEM image of NRFP grouting material (density *ρ* = 0.29 g/cm^3^) with broken micro-foams.

**Figure 5 polymers-15-01493-f005:**
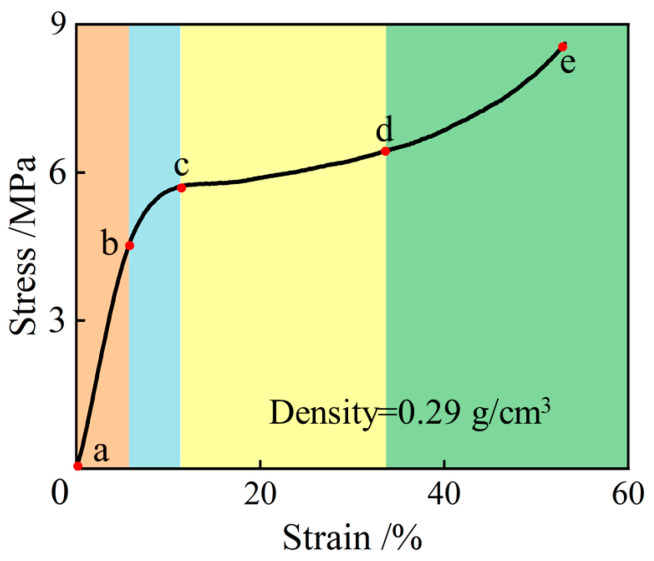
The stress-strain curves of the NRFP sample under uniaxial compression (linear increasing stage (ab), yielding stage (bc), yield plateau stage (cd), and strain hardening stage (de)).

**Figure 6 polymers-15-01493-f006:**
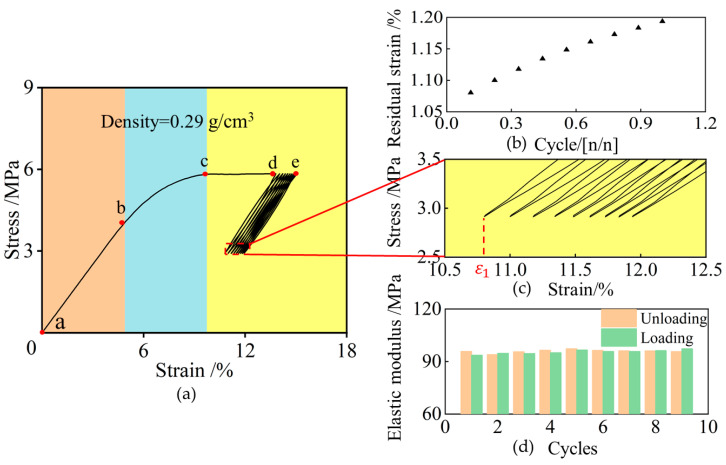
Cyclic loading and unloading experiment results: (**a**) stress-strain curve (linear increasing stage (ab), yielding stage (bc), yield plateau stage (cd), and cyclic loading stage (de)), (**b**) Residual Strain—Normalized Cycle Number, (**c**) local amplification of stress-strain curve, and (**d**) modulus.

**Figure 7 polymers-15-01493-f007:**
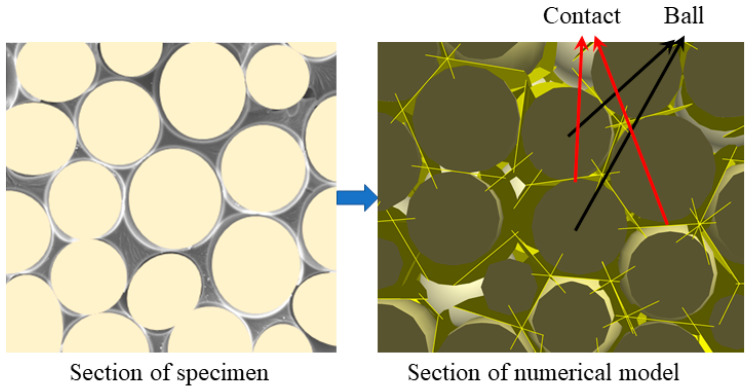
Schematic diagram of mapping the micro-foams in NRFP grouting material to the balls in DEM model.

**Figure 8 polymers-15-01493-f008:**
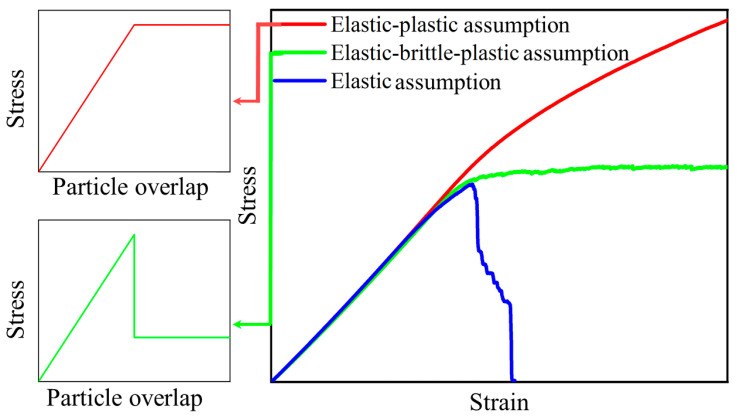
The numerical stress-strain curves were obtained based on different contact models that follow the elastic assumption, elastic-plastic assumption, and elastic-brittle-plastic assumption.

**Figure 9 polymers-15-01493-f009:**
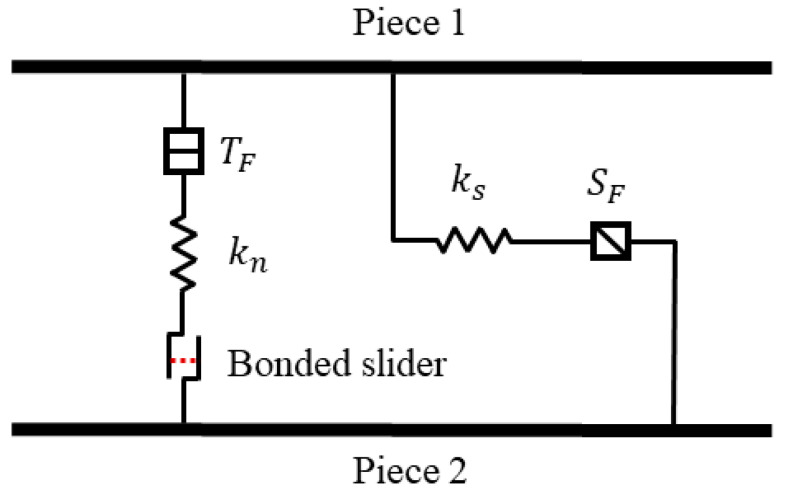
The combination form of the basic elements of the Compression-softening bond model.

**Figure 10 polymers-15-01493-f010:**
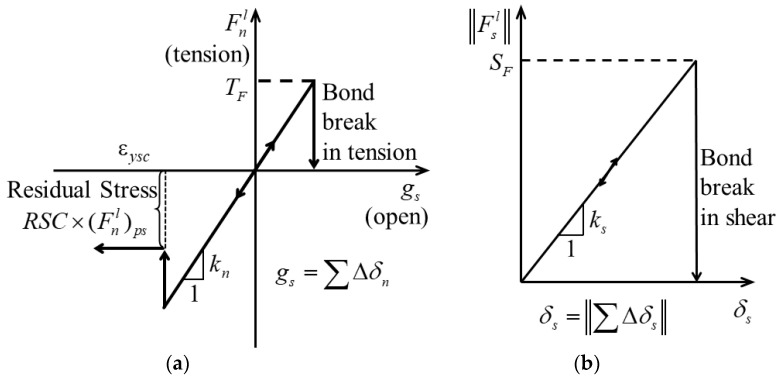
CSB model force-displacement criterion (**a**) normal direction and (**b**) tangential direction.

**Figure 11 polymers-15-01493-f011:**
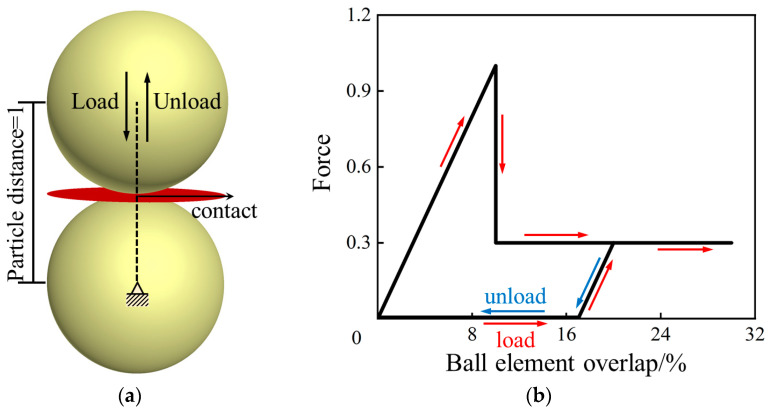
(**a**) PFC model simulating the compression between two balls and (**b**) variation of normal force of the contact element during the simulation process.

**Figure 12 polymers-15-01493-f012:**
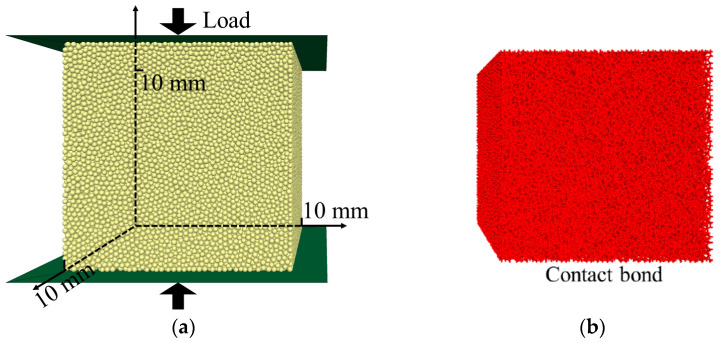
(**a**) PFC model simulating the NRFP sample and the (**b**) contact element constituting the model.

**Figure 13 polymers-15-01493-f013:**
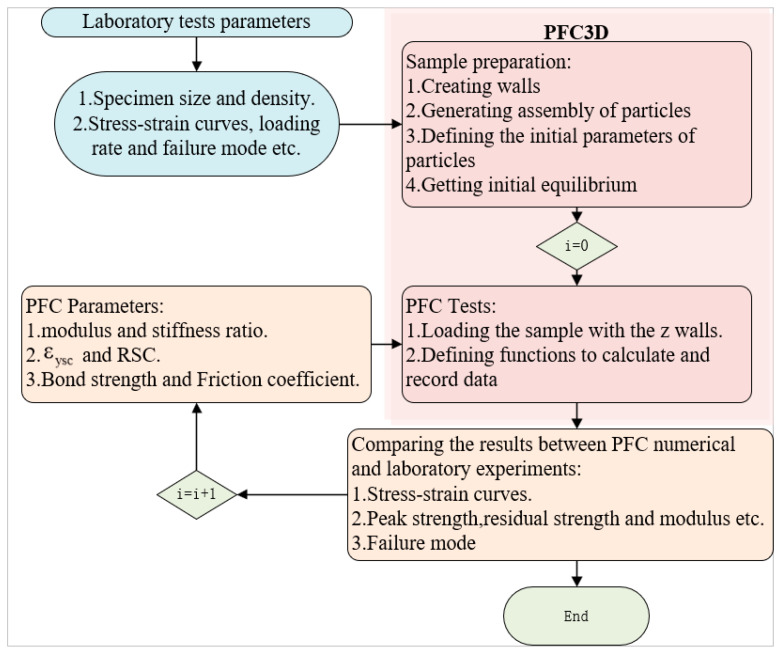
The calibration process of the parameters of the PFC model simulating the NRFP sample.

**Figure 14 polymers-15-01493-f014:**
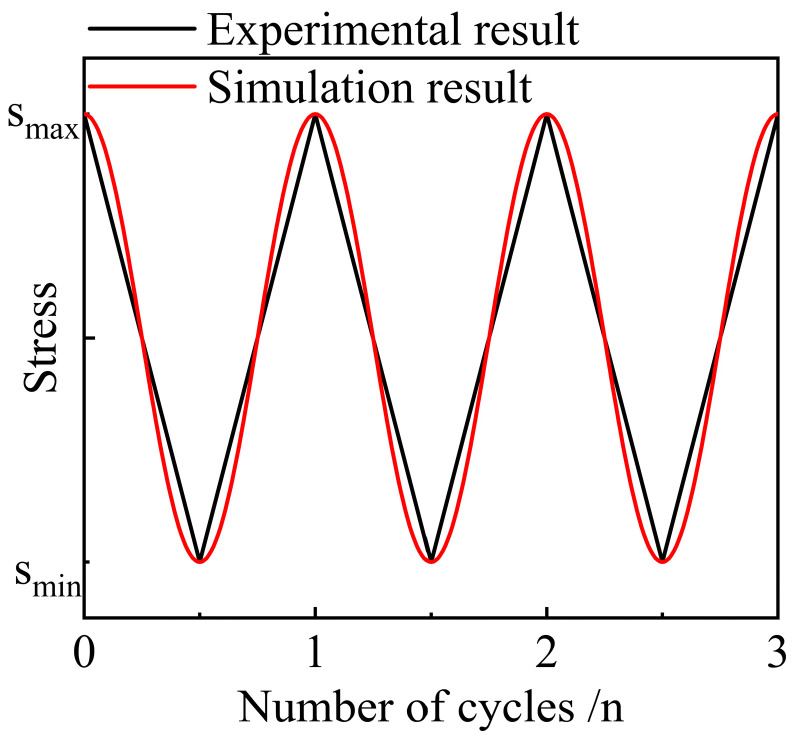
Schematic diagram of loading stress.

**Figure 15 polymers-15-01493-f015:**
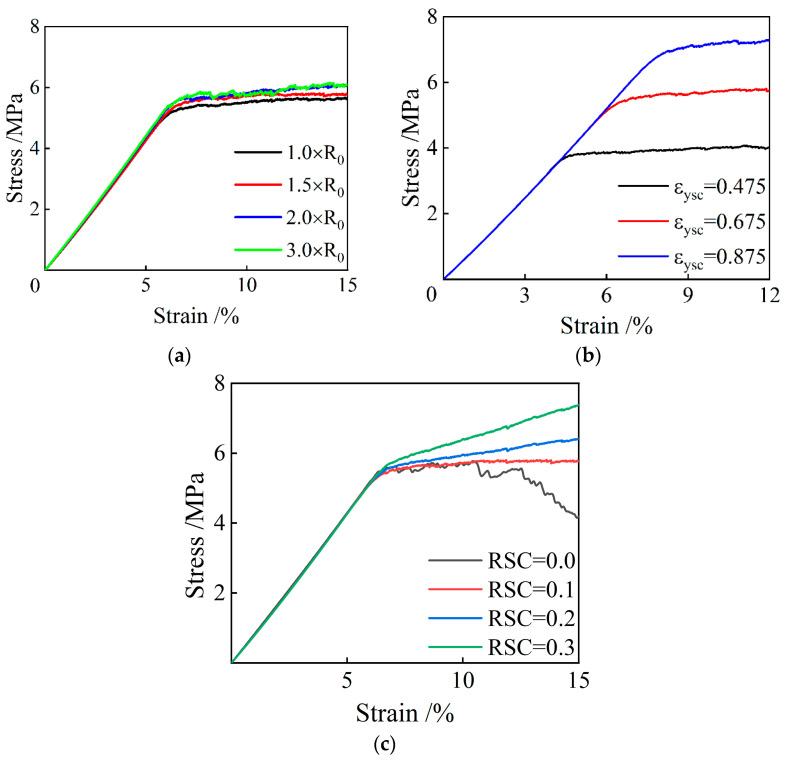
The influences of (**a**) radius of balls, (**b**) yield strain coefficient *ε_ysc_*, and (**c**) residual stress coefficient *RSC* on the stress-strain curves of the PFC models under uniaxial compression.

**Figure 16 polymers-15-01493-f016:**
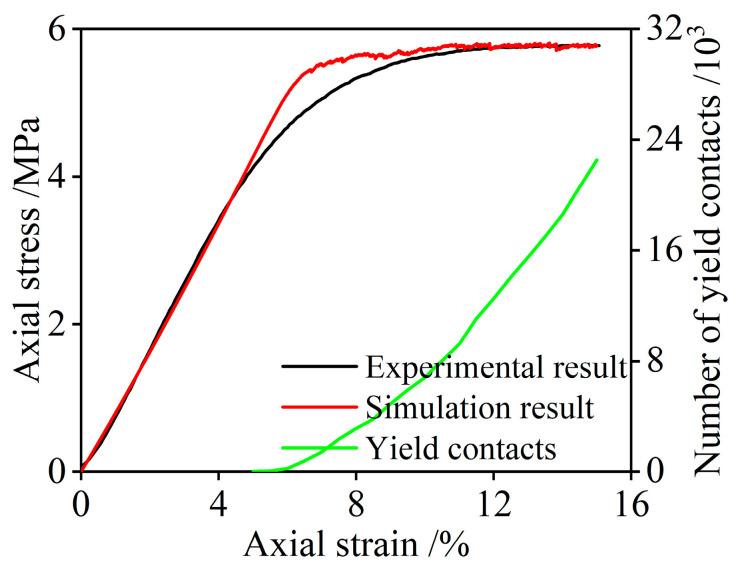
Comparison of the experimental and simulation stress-strain curves and the variations of the number of the yield contact elements with the axial strain.

**Figure 17 polymers-15-01493-f017:**
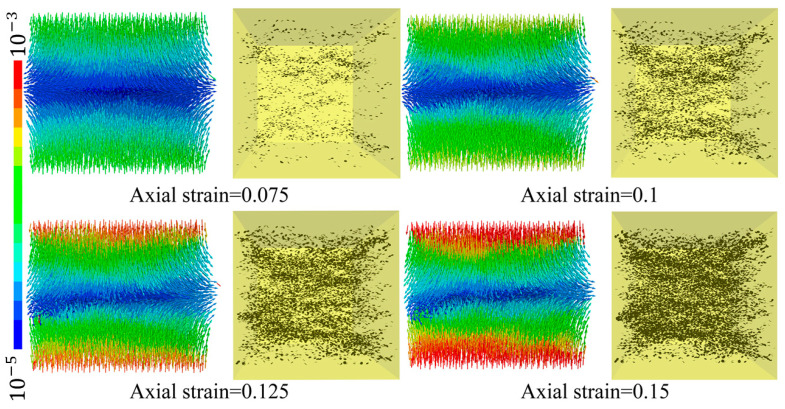
Displacement cloud image and spatial distribution of yield contact.

**Figure 18 polymers-15-01493-f018:**
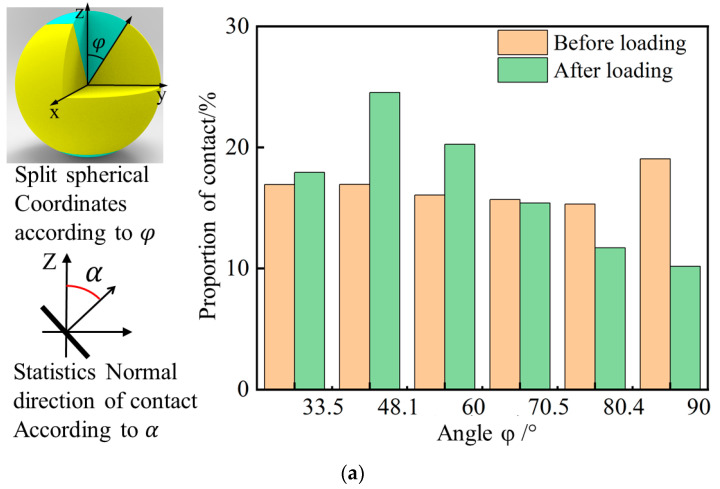
Anisotropy distribution of contacts (**a**) distribution of angle between contact normal and z-axis, (**b**) normal direction distribution of contacts before loading, and (**c**) normal direction distribution of contacts after loading.

**Figure 19 polymers-15-01493-f019:**
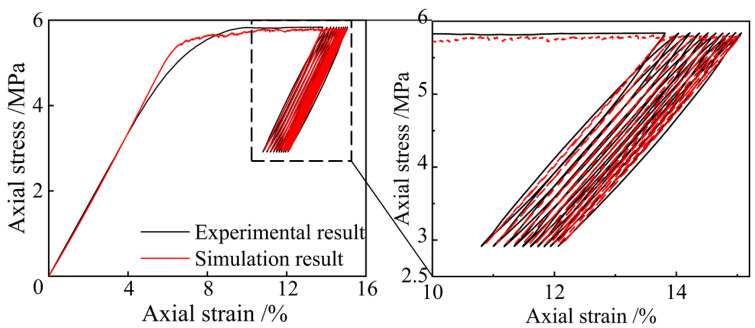
Comparison of the experimental and simulation stress-strain curves of the NRFP sample during the cyclic loading-unloading process.

**Figure 20 polymers-15-01493-f020:**
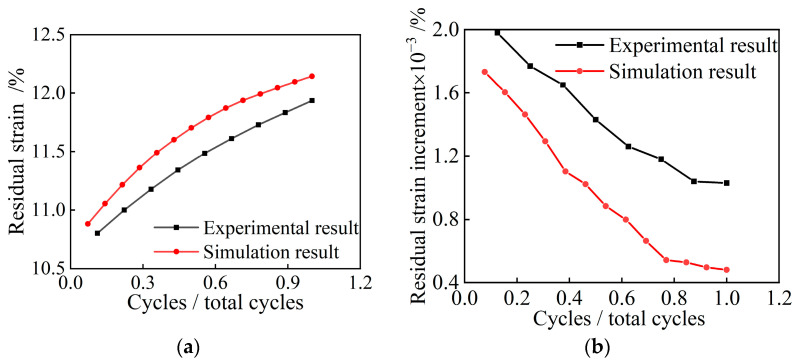
Variations of (**a**) residual strain and (**b**) residual strain increment with the number of loading and unloading cycles.

**Table 1 polymers-15-01493-t001:** General information about the PFC model.

Parameters	Value
Ball radius [μm]	100–125
Porosity [%]	32
Density [kg/m^3^]	290
Edge length of specimen [m]	1 × 10^−2^

**Table 2 polymers-15-01493-t002:** Micro-parameters of CSB model.

Micro-Parameters	Value
modulus [pa]	1.2 × 10^8^
stiffness ratio	2.0
$\varepsilon_{ysc}$	0.475, 0.675, 0.875
*RSC*	0.0, 0.1, 0.2, 0.3
Tensile strength [pa]	1.5 × 10^3^
Shear strength [pa]	1.5 × 10^3^

## Data Availability

All relevant data are within the paper.

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
