# Peer review of "Compression-Softening Bond Model for Non-Water Reactive Foaming Polyurethane Grouting Material"

_polymers, 2023, doi:10.3390/polym15061493_

Round 1
Reviewer 1 Report
Dear Editor
I reviewed the article titled Compression softening bonding model for Non-water Reactive Foaming Polyurethane Grouting Material (Poymers-2259910).
Non-Water Reactive Foaming Polyurethane Joint Filler (NRFP) uniaxial compression tests and cyclic loading tests were performed in this manucript. The suitability of the properties of stage, plateau (plastic flow) stage and densification stage were examined. PFC top units and DLL program have used Via C++ to describe the mechanical response between cell structures. The results of the numerical simulations have validated by comparison with laboratory test results.
This study is an original study in that it contains experimental and simulation results. Acceptable after minor corrections.
-Absract should present the work done better and include the results obtained briefly.
-On line 133, After .... that should start with a lowercase letter after the semicolon, or it is correct to start with a capital letter if replacing the semicolon with a period
-Line 131 Thus …. and Line 209 The elastic …. It has the same typo as on line 133.
-Figures have wrote by abbreviatingfor the text.
-References should be arranged according to the rules of the journal.
The information given in the results and discussion section includes the information that should be given in the conclusion section. The manuscript does not contain a conclusion part.
- If the comments in the Experiments section are transferred to the Results and Discussion section and rearranged, the results and discussion section will be better.
English should be rearranged.
Reviewer 2 Report
1. How the density influences the structure damage of NFRP during loading process? I suggest authors to clarify the same.
2. Which method can be effective for injecting polyurethane in repairing process. I recommend author to mention the same discussion part in manuscript.
3. Authors must check all references are cited inside the manuscript.
4. I suggest authors to incorporate the discussion part regarding the sources for non-water reactive foaming polyurethane.
5. In line no.151, author has mentioned the figure no. 4a. I suggest authors to mention the same in the place of figure presented.
6. Is any relationship between RSC and stress strain curve behavior? I suggest author should confirm the same.
7. Overall manuscript should check for grammatical errors
